# Trastuzumab: More than a Guide in HER2-Positive Cancer Nanomedicine

**DOI:** 10.3390/nano10091674

**Published:** 2020-08-26

**Authors:** Celia Nieto, Milena A. Vega, Eva M. Martín del Valle

**Affiliations:** Departamento de Ingeniería Química, Facultad de Ciencias Químicas, Universidad de Salamanca, 37008 Salamanca, Spain; mvega@usal.es

**Keywords:** HER2 overexpression, trastuzumab, targeted nanoparticles, targeted liposomes, antibody-drug conjugates, conjugation

## Abstract

HER2 overexpression, which occurs in a fifth of diagnosed breast cancers as well as in other types of solid tumors, has been traditionally linked to greater aggressiveness. Nevertheless, the clinical introduction of trastuzumab has helped to improve HER2-positive patients’ outcomes. As a consequence, nanotechnology has taken advantage of the beneficial effects of the administration of this antibody and has employed it to develop HER2-targeting nanomedicines with promising therapeutic activity and limited toxicity. In this review, the molecular pathways that could be responsible for trastuzumab antitumor activity will be briefly summarized. In addition, since the conjugation strategies that are followed to develop targeting nanomedicines are essential to maintaining their efficacy and tolerability, the ones most employed to decorate drug-loaded nanoparticles and liposomes with trastuzumab will be discussed here. Thus, the advantages and disadvantages of performing this trastuzumab conjugation through adsorption or covalent bindings (through carbodiimide, maleimide, and click-chemistry) will be described, and several examples of targeting nanovehicles developed following these strategies will be commented on. Moreover, conjugation methods employed to synthesized trastuzumab-based antibody drug conjugates (ADCs), among which T-DM1 is well known, will be also examined. Finally, although trastuzumab-decorated nanoparticles and liposomes and trastuzumab-based ADCs have proven to have better selectivity and efficacy than loaded drugs, trastuzumab administration is sometimes related to side toxicities and the apparition of resistances. For this reason also, this review focuses at last on the important role that newer antibodies and peptides are acquiring these days in the development of HER2-targeting nanomedicines.

## 1. Introduction

Today, it is well known that cancer is one of the most important public health problems worldwide, since it is the second leading cause of death [1]. Globally, about 1 in 6 deaths is caused by cancer and, in 2018, this complex disease affected almost 20 million people and was responsible for the death of 9.5 million [1,2].

Among the different types of cancer, breast cancer has the second highest incidence, and about 11–12% of the total of new cancer cases that were diagnosed in 2018 were from this tissue [2]. Although there are manifold phenotypes of this disease, approximately 15–20% of breast cancer cases present an overexpression of the human epidermal growth factor receptor-2 (HER2) [3,4], which in addition is also overexpressed in other types of solid tumors [5]. On one hand, the increased expression of this tyrosine kinase receptor is related to cell proliferation, migration, and invasion and, thus, to a poor prognosis for patients and a higher risk of disease recurrence [4,6]. Nevertheless, on the other hand, it has offered the possibility of developing guided-treatment approaches [4], solving one major drawback of conventional chemotherapy: its lack of specificity.

The employment of cytotoxic compounds, either alone or combined with other strategies (surgery or radiotherapy), is the most common first-line treatment against cancer. However, most of these agents exhibit a variable absorption rate and cannot be orally administered. As a consequence, due to its limited effectiveness, chemotherapy must be performed using the systemic route, which is much more uncomfortable for patients. Moreover, since chemotherapy agents are not specifically distributed because of their lack of selectivity, they harm both tumor and normal cells, causing dose-limiting toxicity with severe side effects, such as liver and kidney damage [7,8] (Figure 1). Furthermore, the absence of specificity is also responsible for the apparition of multidrug resistance (MDR) after prolonged exposure to cytotoxic agents, this being one of the most challenging limiting factors of conventional chemotherapy today [7,8,9].

For these reasons, nanotechnology has acquired an essential role during recent years by means of the development of drug delivery systems (DDS), with which it aims to address the downsides of chemotherapy [8]. Thereby, the synthesis of nanomedicines based on viral vectors, drug conjugates. and lipid and polymer nanocarriers has aroused tremendous interest. Among these DDS, nanoparticles (NPs) and liposomes have been preferred for designing nanocarriers due to their advantageous properties. NPs have proven to be more easily soluble in water, increasing the stability and bioavailability of the delivered compounds, and are readily chemically modifiable [10,11]. Therefore, polymeric (chitosan, dextran, pullulan, albumin-based…), ceramic (silica-based or hydroxyapatite), and metallic (mainly gold) NPs have already been used as platforms for the development of new DDS [12,13,14,15]. Similarly, liposomes have been shown to be biocompatible, have little toxicity, and are capable of promoting a controlled drug release [16].

Thanks to the characteristic properties of cancer tissues, there are two mechanisms by which NPs and liposomes can deliver drugs to tumors: passive targeting and active targeting [17]. The passive accumulation process is a direct consequence of the enhanced permeability and retention (EPR) effect, induced by the leaky blood vasculature and impaired lymphatic system of the solid tumors. NPs with an appropriate size (100–400 nm) and superficial charge (preferably negative) can achieve lengthy blood circulation times and accumulate at tumoral sites by diffusion and convection processes. Nonetheless, this passive delivery has its own issues, since the EPR effect does not occur in every tumor (as in those in which there is a great hypoxia), and vessel permeability is not usually homogeneous [8,17,18]. Besides, passive targeting can promote MDR apparition due to its lack of control and the consequent undue drug accumulation in cancer cells, a fact which, on the contrary, can be overcome by active targeting [17]. This second delivery process accomplishes a specific recognition of tumor cells that favors controlled DDS internalization in them, increasing their therapeutic efficiency [19]. Such targeting can be attained by chemical changes in the NP and liposome surface, making it more reactive to the tumor microenvironment, or with ligands that specifically recognize overexpressed receptors or proteins in cancer cells, such as several peptides, aptamers, and antibodies [17,18,19,20]. Among these three targeting molecules, peptides and aptamers are smaller, less immunogenic, more easily chemically modifiable, and more temperature-stable than antibodies. However, antibody properties have been more extensively characterized, making them indispensable for cancer research, diagnosis, and therapy [21,22].

Furthermore, aside from the synthesis of DDS, antibodies have also aroused interest in cancer nanomedicine because of the development of antibody-drug conjugates (ADCs), which have also been designed to increase the efficacy of conventional chemotherapy [23,24].

Among these antibodies, the most commonly used in the treatment against HER2+ tumors are pertuzumab and trastuzumab, two recombinant humanized monoclonal antibodies [25,26]. Of these, trastuzumab (Herceptin^TM^, Genentech/Roche, San Francisco, FL, USA) (Tmab) specifically binds to the extracellular domain IV of the aforementioned HER2 and has allowed the survival rate of patients who suffer from HER2-positive (HER2+) breast cancer to increase [6]. For this reason, this antibody has been extensively used for both the development of guided NPs and liposomes and for the development of ADCs, and is the protein on which this review study is focused (Figure 1).

Thereby, this article summarizes how the clinical employment of Tmab has helped to improve the outcome of patients who suffer from HER2+ breast cancer, as well as the molecular mechanisms that could be behind its antineoplastic activity. Moreover, the main covalent and non-covalent strategies that are followed in the nanomedicine field to decorate NPs and liposomes with this antibody have been examined in detail, along with the ones employed to synthesize Tmab-based ADCs. At the end, the future role of Tmab in the development of novel anti-HER2 nanomedicines is also discussed.

## 2. Trastuzumab: More Than a Guide for Nanomedicines

The clinical utilization of Tmab was firstly approved by the US Food and Drug Administration (FDA) and the European Medicines Agency (EMA) in 1998 and in 2000, respectively, to treat patients with HER2+ metastatic breast cancer (MBC). Several years later (in 2006 and 2011), these two organisms also authorized the employment of this recombinant antibody as adjuvant therapy for patients with HER2+ early breast cancer (EBC) and, finally, in 2015 Tmab was added to the Essential Medicines List of the World Health Organization (WHO) [27]. Such addition was the outcome of the beneficial effects that Tmab has proven to have for women with HER2+ MBC and EBC when it is administered with chemotherapy, slowing down tumor progression, inducing tumor regression, and increasing patients’ overall survival rate [28]. For instance, when Tmab is administered in the first-line treatment of MBC, it induces tumor regression in 30–35% of patients and increases patients’ disease-free survival rate after five years by 10% in comparison with only the administration of conventional cytotoxic drugs [28,29]. Moreover, Tmab is able to reduce their risk of disease recurrence by 50% when it is given to patients who suffer from EBC, too [28,30].

However, the molecular pathways that are behind these positive results are not completely known and remain an active research area [31]. Among them, three main mechanisms have been proposed to be responsible for Tmab antitumor activity: (I) cell cycle arrest triggered by the inhibition of the MAPK and PI3K signaling cascades; (II) antibody-dependent cellular cytotoxicity (ADCC); and (III) the increased production of anti-angiogenic factors (Figure 2) [31,32]. The aberrant activation of the PI3K/AKT/mTOR and MEK/ERK pathways has been linked to an induction of cellular proliferation and survival rate, while the in vivo inhibition of such pathways has shown to reduce tumor growth [33]. In this way, since Tmab administration inhibits transforming signals downstream of HER2, this antibody can trigger cell cycle arrest and induce cancer cell apoptosis (Figure 2a) [34]. Otherwise, targeted cells opsonized by immunoglobulin (IgG)-based monoclonal antibodies, such as Tmab, are able to bind and activate FcɣR-bearing immune effector cells, like the NK cells, and this fact results in a target cell lysis (Figure 2b) [35]. Finally, it has also been demonstrated that, through the mentioned inhibition of the PI3K/AKT/mTOR pathway, Tmab decreases the expression of some proangiogenic factors like the vascular endothelial growth factor (VEGF) and interleukin-8 (IL-8) (Figure 2c) [36].

Thereby, such three mechanisms could be implicated in the Tmab-mediated active targeting of NPs and liposomes decorated by this antibody on their surface and of Tmab-based ADCs, achieving not only a guided treatment but also a synergy between Tmab and the delivered drugs. For this reason and also because Tmab has been used in the clinic for almost two decades with the approval of the main drug regulatory agencies, a large number of NPs and liposomes have been already conjugated to this antibody to improve conventional chemotherapy efficacy, and even a Tmab-ADC is already being commercialized [23,37].

## 3. Nanoparticle and Liposome Functionalization with Trastuzumab: Usual Strategies

The different strategies that have been pursued to anchor Tmab to the surface of sundry types of NPs and liposomes can be classified into two main groups, depending on whether or not the antibody-binding is covalent. In this way, works in which Tmab has been adsorbed on several particulate nanosystems can be found in the literature, while in other studies the antibody has been covalently attached to NPs and liposomes following several strategies, such as with carbodiimide, maleimide, or click-chemistry [17,38].

### 3.1. Functionalization through Trastuzumab Adsorption

Adsorption immobilization comprises physical and ionic bindings. In the first sort of adsorption, electrostatic and hydrophobic interactions as well as the hydrogen binding are comprised. On the other side, ionic binding occurs when the antibody and the NP surface have opposite charges (Figure 3a). In any case, both adsorptions are reversible and the non-covalent functionalization with them is rapid and simple, since it does not require any chemical modification [17]. Taking advantage of physical adsorption, Liu et al. [39] developed polylactide-d-α-tocopheryl polyethylene glycol succinate (PLA-TPGS) NPs loaded with docetaxel and decorated by Tmab, and demonstrated that a synergist effect could be achieved when the drug and the antibody were simultaneously administered. In the same manner and thanks to electrostatic interactions, Yu et al. [40] and Zhang et al. [41] functionalized with Tmab the surface of polyethylenimine/poly(lactic-co-glycolic acid) (PEI/PLGA) NPs that transported paclitaxel and docetaxel, respectively, to specifically treat HER2+ breast cancer cells. Likewise, Sun et al. [42] adsorbed Herceptin^TM^ to their PLGA/montmorillonite (MTT) NPs, which also carried paclitaxel to breast tumor cells overexpressing HER2.

Nevertheless, DDS functionalization through adsorption methods is less stable than covalent bindings and requires high amounts of antibodies, which makes the conjugation process more expensive. Besides this, the adsorbed antibody can suffer conformational changes that would decrease its antigen recognition capacity and make the functionalization process less reproducible (Figure 3b) [17]. This fact was precisely demonstrated by Choi et al. [43], who compared how the employed Herceptin^TM^- functionalization method conditioned the antiproliferative activity of docetaxel-PLGA NPs. These authors modified the mentioned NP surface with Tmab through adsorption, charged adsorption, and bio-conjugation. As a result, they found that this last covalent conjugation process was more efficient and that bio-conjugated NPs had a greater stability, cell internalization rate, and cytotoxicity than the NPs that were functionalized through the adsorption processes [43].

### 3.2. Trastuzumab-Functionalization through Covalent Bindings

As it has been before mentioned, Tmab covalent binding can be achieved with different strategies, the most common ones being carbodiimide chemistry, maleimide chemistry, and click-chemistry.

The first one, carbodiimide chemistry [44], is probably the most used one. It requires the employment of 1-ethyl-3-(3-dimethylaminopropyl) cardodiimide (EDC), a zero-length crosslinking agent that allows the binding of the carboxyl groups present in the DDS surface with the primary amine functional groups of antibodies. When EDC reacts in one-step with carboxyl groups, O-acylisourea esters, which are highly reactive, are generated. Then, such intermediate compounds react with primary amines, and amide bonds are finally formed. The inconvenient aspect is that O-acylisourea esters are not stable enough, and intra- and inter-molecular bindings can take place between antibody functional groups. In order to avoid this fact, N-hydroxysulfoxuccinimide (NHS) is usually incorporated in the carbodiimide chemistry, although its use is not mandatory [45]. With its addition, the reaction takes place in two steps with an increased efficiency, since O-acylisourea esters become semi-stable esters (Figure 4a) [17,46].

The main advantage of this sort of covalent functionalization, which explains its large use, is its simplicity. Primary amine groups are abundant in the antibody surface and no chemical modification of the NPs is required to carry it out [17]. Because of that, carbodiimide chemistry has been widely utilized in the literature to anchor Tmab to different particle nanosystems. For instance, Choi et al. chose it to make the comparative study that was aforementioned [42]. Similarly, Zhou et al. [47], Mehata et al. [6], and Nieto et al. [48] decorated the surface of their synthesized NPs with Tmab by taking advantage of this method to treat HER2+ breast carcinoma cells. The first authors [47] developed PLGA-poly-l-histidine (Phis)-polyethylene glycol (PEG) NPs and loaded them with doxorubicin. Mehata et al. [6] obtained TPGS-g-chitosan NPs that carried docetaxel to the mentioned cells and, finally, Nieto el al. [48] synthesized alginate-piperazine NPs to improve and guide paclitaxel treatment. In addition, these last authors also proved that their nanosystem was able to reduce the rate of survival of other types of cancer cells that also overexpress HER2, such as ovarian tumor cells. In the same way, Domínguez-Ríos et al. [49] conjugated Tmab to the surface of PLGA NPs to treat a HER2+ ovarian cancer cell line, and Arya et al. [50] did this with chitosan NPs to treat HER2+ pancreatic tumors. Furthermore, carbodiimide chemistry has not only been used to create Herceptin^TM^-DDS, but also to develop NPs with which the bio-separation, selective radiotherapy, and hyperthermia of HER2+ cancer cells could be performed [51,52,53].

As it can be checked, carbodiimide chemistry has been employed to modify the surface of NPs of a very different composition, but it presents a handicap. Carbodiimide chemistry is not a really selective coupling method, since primary amine groups can be found anywhere in the antibody surface. Thus, it lacks control of antibody orientation, and as consequence other covalent functionalization methods which are more site-specific have been preferred by other researchers [17].

Functionalization performed with maleimide chemistry, which is based on bindings through the sulfhydryl or thiol groups of antibodies, is one of such methods [54]. These groups are less abundant on the antibody surface than primary amines, and generally they are oxidized and present in form of disulfide bonds that couple pairs of cysteines. These amino acids, which are essential for the formation of the tertiary or quaternary structure of proteins, are the most reactive nucleophiles in them. Nonetheless, antibody conjugation only can occur through free or reduced sulfhydryl groups, which have to be introduced. This introduction can be achieved through reaction with primary amines or through the reduction of disulfide bonds, which can be cleaved with different reducing agents, such as dithiothreitol (DTT) or 2-Mercaptoethanol (β-ME). Once obtained, reactive sulfhydryl groups can react towards maleimide, α-haloacetyls, and pyridyl disulfides [17,55]. With the first two compounds, an irreversible thioether linkage is formed, but maleimide-activate crosslinkers present a higher selectivity for the sulfhydryl side chain of cysteines and more rapid ligation kinetics in aqueous conditions and have received more attention [17,56] (Figure 4b). Thus, for such maleimide activation, two different strategies can be pursued: (i) DDS functionalization, introducing thiol groups or maleimides; or (ii) the employment of hetero- or homobifunctional linkers with one or two maleimides at the ends, respectively. Both strategies have been followed in order to conjugate NPs and liposomes with Tmab.

For example, the first option was the one followed by Taheri et al. in their study [57]. These authors conjugated methotrexate (MTX) to human serum albumin (HSA) and, after the crosslinking such protein, they obtained NPs that were decorated by Tmab to treat HER2+ breast cancer. To achieve this decoration, Taheri et al. introduced thiol functional groups in the NPs that they synthesized, and activated the anti-HER2 antibody with 4-maleimidobutyric acid-N hydroxysuccinimide (GMBS). After allowing them to react, they obtained covalent Tmab-attached MTX-HSA NPs [57]. In addition, this strategy was the one employed by Nguyen et al. and Amin et al., too [58,59], to attach Tmab to liposomes. All these authors synthesized liposomes with a maleimide-terminated PEG lipid conjugate (DSPE-PEG-Mal) and thiolated Tmab in order to perform a covalent conjugation. Thereby, Nguyen et al. managed to develop PEGylated liposomes in which they included rapamycin and polypyrrole (PPɣ) NPs for the targeted chemo-photothermal therapy of HER2+ breast cancer cells [58], while Amin et al. created Tmab-conjugated liposomes to specifically deliver idarubicin to the same sort of tumor cells [59].

On the other hand, the second option was chosen in works such as those of Chiang et al. [60], Jang et al. [61], Kesavan et al. [62], Steinhauser et al. [63], and Kubota et al. [64]. The first authors, in order to trigger the same type of tumors as the previous researchers [57,58,59], developed double emulsion nanocapsules (DENCs) in which they simultaneously encapsulated paclitaxel and doxorubicin. Then, on their surface they attached a magnetic targeting and Tmab in order to achieve a combined therapy and, for the antibody conjugation, followed the succinimidyl-4-(*N*-maleimidomethyl)-cyclohexane-1-carboxylate (SMCC) method. Thereby, they carried out a thiol-functionalization of their nanocapsules and a maleimide-activation of the antibody with the SMCC Pierce^TM^, a heterobifunctional crosslinker that contains NHS ester and maleimide groups [60]. Similarly, Jang et al. prepared liposomes encapsulating silica-core-shell magnetic NPs and attached Tmab on their surface through the SMCC method to treat HER2+ breast tumors using magnetic resonance imaging (MRI) monitoring. Briefly, these authors formed an amide bond from the primary amine of their particles employing sulfo-SMCC and, later, thiolated Tmab with Traut’s reagent, proving that their conjugated liposomes accumulated in detectable amounts in tumors overexpressing HER2 [61]. In the same manner, Kesavan et al. introduced maleimide groups in Tmab and attached such antibodies to the surface of polyamidoamine dendrimer–cisplatin NPs that had been also functionalized with diglycolamic acid to treat HER2-overexpressing ovarian tumor cells. To achieve this goal, Kesavan et al. conjugated the amine groups of their NPs with LC-SPDP, another heterobifunctional crosslinker, and carried out a reduction reaction with DTT to obtain reactive thiol groups on the surface of the NPs that could react with Tmab-maleimide groups [62]. Just the opposite, Steinhauser et al. performed a thiolation of Tmab with the use of 2-iminothiolane and activated the HSA NPs that they had been previously obtained with a heterobifunctional crosslinker (NHS-PEG5000-Mal) with similar terminal functional groups to those of SMCC Pierce^TM^. In this way, these authors were able to develop a HER2-guided drug carrier system [63]. Finally, Kubota et al. synthesized gold NPs decorated with Tmab to treat HER2+ gastric cancer cells that were resistant to this antibody [64]. To anchor Tmab to their surface, these authors employed a linker that consisted of a short PEG chain terminated at one end by a hydrazide moiety and at the other end by two thiol groups [65], and added methoxyPEG-SH to cap any remaining bare surfaces of the gold NPs [64].

Although conferring site-specific conjugation to cysteine residues, NP and liposome functionalization through maleimide chemistry also has proven disadvantages: maleimide can react with thiol groups present in serum proteins (like albumin) and, as a consequence, resulting bioconjugates have been shown to be inherently unstable in vivo. In addition, this strategy involves the loss of a covalent bond between the antibody chains. As a solution, other forming antibody-NP/liposome conjugates strategies, such as click-chemistry, have appeared [17,66].

The click-chemistry term, which was firstly proposed by Kolb et al. in 2001 [67], refers to a group of powerful chemical reactions which are orthogonal with other functional groups (amines, thiols, carboxylic acids…), simple to perform, favorable in aqueous conditions, with high yields, and that generate minimal byproducts [68,69,70]. The first reaction that was called click-chemistry, which is the most widely used today in nanomedicine, was the copper-catalyzed cycloaddition between azides and alkynes that generates 1,2,3-triazoles (CuAAC). Later, cycloadditions between strain-promoted alkynes and azides (SPAAC) enabled copper-free click-chemistry and started to be preferred to CuAAC to functionalize DDS to prevent copper bioaccumulation [68]. In the end, the inverse-demand Diels–Alder reaction with 1,2,4,5-tetrazine (Tz) and trans-cyclooctene (TCO) (iEDDA) provided an ungraded reaction rate and also began to be applied in biomedicine (Figure 5) [68,71]. As examples of the application of the click-chemistry in the development of Tmab-nanoconjugates, the works of Greene et al. [66], Yoo et al. [72], and Keinänen et al. [73] can be highlighted.

Foremost, Greene et al. described in an anterior study a new approach to insert pyridazinedione moieties bearing reactive handles into antibody-reduced disulfide bonds for enabling the incorporation of click-domains without losing covalent linkages between the antibody chains [74]. Then, in the work quoted here, these authors took advantage of such an approach to site-selectively modify the F(ab) domain of Tmab to bear a strained alkyne handle distal to the paratope and to conjugate it to azide-functionalized PLGA NPs. For such a purpose, they incorporated a complementary azide moiety into the NPs and synthesized a heterobifunctional linker to conjugate the Tmab-F(ab) disulfide to them. In such linker, Greene et al. included a strained alkyne bicyclononyne (BCN) and employed SPAAC to develop guided NPs for the treatment of HER2+ breast cancer cells, showing that the click-chemistry that they used was more efficient than the NHS ester one [66]. Secondly, Yoo et al. chose the inverse-demand Diels–Alder reaction between Tz and TCO to perform a two-step treatment of HER2+ cancer cells with Tmab and liposomes that had been loaded with the anticancer drug SN38. They performed the conjugation of Tmab with TCO by means of a linker (TCO-PEG4-NHS ester) and modified the surface of their liposomes with the Tz groups. Later, these authors treated cancer cells with the TCO-modified Tmab, allowing the TCO groups to remain exposed on the tumor cell surface, and performed the second step of the treatment with the Tz-modified SN38-liposomes, which bound the TCO groups via click-chemistry, achieving a chemotherapy enhancement [72]. Finally, Keinänen et al. also employed the same iEDDA methodology to Yoo et al., but with a different aim: in vivo tracing the internalization of Tmab with a fluorine-18 labelled-Tz tracer. Thus, they modified the antibody with TCO and injected it in mice with HER2+ breast cancer tumors, and successfully visualized it by positron emission tomography (PET) imaging [73].

A summary of all the Tmab-guided DDS, developed following the different non-covalent and covalent strategies explained here, can be found in Table 1.

## 4. Trastuzumab Role in Antibody-Drug Conjugates Development

As reported in the introduction section, antibodies have also become relevant in nanomedicine thanks to the development of ADCs. These glycoproteins present an insufficient clinical activity themselves and ADCs emerged to empower their antiproliferative effect [75,76]. Thereby, such conjugates are produced with the objective of selectively ablating cancer cells by combining the action of a highly potent cytotoxic compound with antibody specificity for a target antigen, with these two compounds being conjugated through a linker [24,75]. In this way, after binding it, the ADC-antigen complex is internalized and transported to cellular organelles (generally lysosomes) where the release of the attached drug can take place [75]. To improve its therapeutic activity without compromising safety, ADCs must limit the exposure of normal tissues to the transported drugs and only deliver the payload to the tumor cells that express the chosen antigen [75,76]. For this reason, linker technologies that ensure an adequate stability of the drug in ADCs are required so that the drug release does not occur in circulation [76]. Besides, the method employed in ADC conjugation conditions the drug loading stoichiometry and homogeneity and determines its anti-tumor activity, efficacy, and tolerability [76,77].

For such conjugations, at the beginning of ADC development, acid-labile hydrazone linkers that can be cleaved in the acid environment of endosomes and thus allow the release of the ADC payload in these organelles were selected. However, disulfide-based linkers demonstrated later to be a better choice because they were more stable at a physiological pH, and nowadays they are normally preferred to anchor cytotoxic compounds to antibodies [75]. When they are employed, the conjugation of linker drugs to an antibody occurs at accessible reactive amino acids derived from the reduction of its interchain disulfide bonds, and three main methods for achieving such accessibility can be distinguished: (i) the acylation of lysines, (ii) the alkylation of the reduced interchain-disulfides of cysteines, and (iii) the alkylation of genetically engineered cysteines [75,76,78].

In the majority of ADCs that have been developed and are in clinical trials, drug molecules have been covalently bound through lysine and cysteine linkers, following the first two aforementioned strategies (Figure 6a) [24]. Between both of them, the alkylation of reduced interchain-disulfides of cysteines has been normally chosen, since there are more much lysines present in the antibody surface than interchain-cysteines (40 lysines per antibody versus 8 exposed cysteine sulfhydryl groups), and the heterogeneity of the reaction is reduced when cysteines are selected to anchor the cytotoxic compounds [24,76,78]. In any case, maleimide chemistry is usually chosen to synthesize both types of disulfide-based ADCs, and the employed linkers can be either cleavable or non-cleavable.

The first ones, the cleavable linkers, include an engineered lysosomal specific protease or are disulfide-bond-based glutathione (GSH)-sensitive, since the intracellular concentration of this molecule is much higher than it is in circulation. Otherwise, non-cleavable thioether linkers are those that make necessary a post-internalization degradation in lysosomes of the ADC to release the payload (Figure 6b) [24]. They have a better stability in the bloodstream and longer half-lives and, hence, a smaller risk of off-target toxicity than that of cleavable linkers [24,79]. For this reason, this sort of linker was the one that was employed to synthesize ado-trastuzumab emtansine (T-DM1) [80], which was the first anti-HER2 ADC that started to be commercialized in 2013 (Kadcyla^®^) [23].

Thus, T-DM1 is integrated by a non-cleavable linker that allows the attachment of a derivative of maytansine (DM1) to Tmab [24,81]. Maytansine, which is a natural inhibitor of tubulin polymerization, was selected to be part of this ADC because it has a great stability and an appropriate aqueous solubility. In addition, it was shown to be orders of magnitude more potent than other clinically used anticancer drugs. Notwithstanding, although natural maytansine has proper biological and biochemical properties, it lacks a suitable functional group to be conjugated to an antibody, and a thiol group had to be introduced in its structure [82,83]. Structure activity relationships (SAR) studies were carried out in order to determine the most proper modification site to avoid an alteration of maytansine potency, and in the end a thiol group was introduced in the aminoacyl side chain C3 of the drug. Then, the heterobifunctional crosslinking agent SMCC was chosen to attach DM1 to Tmab through its lysine residues by means of the formation of a thioether binding [82]. As a result, an average of 3.5 DM1 molecules were linked per antibody, and the resulting conjugate maintained good biochemical properties [82,83]. After extensive preclinical and clinical evaluations of its biological activity, pharmacokinetics, metabolism, and tolerability, the FDA finally approved T-DM1 administration seven years ago to treat patients with HER2+ MBC, previously treated with Tmab and taxanes [82]. Furthermore, the encouraging results that were obtained during the T-DM1 evaluations have caused the development of novel ADCs in which the antibody Tmab has been maintained, but different cytotoxic drugs have been coupled to it through different linkers [82,83]. Several examples can be found in the works of Xu et al. [83], Robinson et al. [84,85,86], and Shen et al. [77].

The review study of Xu et al. is focused in the main properties of two different Tmab-based ADCs, SYD98 and Tmab deruxtecan (DS-8201a), which were developed with the purpose of reducing T-DM1 resistance and improving its efficacy in heterogeneous tumors [83]. The first one, SYD98, was synthesized by Elgersma et al. in 2015 [87], where they chose a duocarmycin derivative (Seco-DUBA) that has a better solubility and stability than the parent alkylating drug to be attached to Tmab. For such an attachment, they selected a peptide linker that conjugated Seco-DUBA molecules to Tmab through the hydroxyl groups present in their DNA-alkylating moiety. This linker, unlike the T-DM1 one, can be cleavable by cathepsin B, a lysosome cysteine-protease present in cells [83,87]. Similarly, Ogitani et al., who obtained DS-8201a in 2016 [88], also preferred a cleavable linker to be part of their Tmab-based ADC. These authors linked a maleic acid to Tmab and, through it, joined the commercially available linker BOC-GGFG-OH that is selectively cleaved by lysozymes. As a cytotoxic drug, Ogitani et al. decided to employ a camptothecin derivative (DXd) that was developed to improve the solubility and biological activity of the original camptothecin, a topoisomerase poison [83,88]. Both of them, SYD98 and DS-8201a, are now in clinical trials in which their suitability for the treatment of HER2+ breast, gastric, and lung cancers is being evaluated with promising results [83].

Otherwise, Robinson and co-workers demonstrated that site-selective disulfide bridging with small molecules, such as next-generation maleimides (NGMs) [84,86] and pyridazinediones (PDs) [85], constitutes a proper conjugation strategy to develop stable ADCs. In order to attach the potent anticancer drug monomethyl auristatin E (MMAE) to Tmab, they reduced the antibody native interchain disulfide-bonds with tris-2-carboxyethylphosphine (TCEP). Next, they performed a functional re-bridging with either an NGM or a PD molecule and conjugated NGM-MMAE and PD-MMAE to Tmab, obtaining efficient ADCs [84,85]. In addition, in an anterior study these authors followed the same strategy with NGMs to synthesize a Tmab-ADC with loaded doxorubicin, anticipating that the NGM platform could have considerable utility for the development of ADCs [86].

At last, Shen et al. also built Tmab-ADCs loaded with MMAE but, to assess the impact of the conjugation site, they engineered cysteines at three different Tmab sites, differing in solvent accessibility and local charge. Once obtained, they attached MMAE to them through maleimide chemistry with a maleimido-caproyl-valine-citruline-p-amino-benzyloxy carbonyl (MC-vc-PAB) linker, and showed not only the linker choice conditions ADC biological activity, but also the conjugation site [77].

Since, in some studies, it has been shown that the location of attached compounds is not as relevant as their stoichiometry and that heavily loaded conjugates are quickly removed from the circulation, recombinant methods have begun to acquire more importance in this nanotechnology field [24,76]. ADC conjugation through the alkylation of genetically engineered cysteines arose for this reason. Mentioned above, it is the most recent strategy to attach cytotoxic compounds to antibodies and is based on protein-engineering alterations that allow the binding of a particular number of drug molecules per ADC. Antibody modifications can be performed through enzymatic conjugation and through the insertion of reactive cysteines or chemoselective functional groups of unnatural amino acids in its structure, but there are still many challenges concerning these approaches, and any ADC developed following them has reached the clinic yet [78].

## 5. Current and Future Situation of Trastuzumab-Based Nanomedicine

As has been stated, conventional adjuvant chemotherapy with Tmab results in a significant prolongation of disease-free and overall survival rates and has revolutionized the treatment of HER2+ breast cancer [89,90]. As consequence and since HER2 is not only overexpressed in this type of cancer, many studies in the preliminary stages have proposed the administration of this adjuvant therapy to treat other types of HER2+ tumors, including ovarian, bladder, and lung ones [91]. In this way, the Tmab-decorated NPs developed to target these solid tumors other than breast tumors can be already found in the literature [48,49,50,62,64].

However, the clinical administration of Tmab does not only have advantages. Some adverse effects, such as gastrointestinal and pulmonary symptoms, hematologic deficiencies, and especially cardiac toxicity, have been linked to Tmab use. Moreover, between 15 and 25% of the patients who have received Tmab therapy experience disease recurrence [90,92]. Manifold mechanisms of primary and treatment-emergent resistance to this antibody have been purposed, including compensatory signaling from either HER family members or other receptor types (such as epidermal growth factor or vascular endothelial growth factor receptors (EGFR and VEGFR)) [92]. For this reason, other antibodies and peptides that are able to block HER2 dimerization with other HER family members or that inhibit simultaneously other receptors have been developed [31]. Among them, two antibodies have been already approved by the FDA to treat HER2+ advanced breast cancer: pertuzumab (Perjeta^®^) and lapatinib (Tykerb^TM^) [31,90]. On one hand, pertuzumab is a humanized recombinant antibody that interferes with the HER3-dimerization domain of HER2, inhibiting cancer cell proliferation by blocking the HER3-dependent signaling pathway. Even though this antibody has shown a modest anti-HER2 efficacy when it is administered alone, it has been demonstrated to have a synergistic effect with Tmab. Because of this, the FDA approved its utilization, in combination with Tmab and docetaxel, to treat HER2+ MBC, and new regimens are being studied to improve the pertuzumab efficacy and toxicity. On the other hand, lapatinib is the single tyrosine-kinase inhibitor (TKI) whose use has been approved to block HER2 and EGFR receptors together. It presents a great cancer inhibitory effect and enhances Tmab activity, too [31,90]. In addition, apart from these two antibodies whose clinical employment is already permitted, many other anti-HER2 antibodies have been developed and are now in the advanced stages of clinical trials: neratinib, pyronitib, afatinib, pazopanib, ertumaxomab, hertuzumab, etc. [5,31,90].

With the growth of novel antibodies and peptides that target HER2, the development of new guided NPs, liposomes, and ADCs conjugated to them has also been propitiated. Regarding NPs, studies of different authors who have encapsulated lapatinib in them can be easily found. For example, Wan et al. and Zhang et al. [93,94], taking into account the high binding efficiency of lapatinib to HSA, obtained NPs based on this protein that they functionalized with the mentioned antibody to increase its low aqueous solubility. Both nanosystems were able to inhibit HER2+ breast cancer cell proliferation. In a similar manner, Mobaserri et al. [95], with the aim of improving lapatinib solubility and bioavailability, encapsulated it in dextran-chitosan NPs that were also demonstrated to have an anti-HER2+ cell growth activity. On the other hand and regarding liposomes, Singh et al. synthesized chitosan-modified liposomes and decorated them with an anti-HER2 tumor homing peptide (THP) (WNLPWYYSVSPTC) to specifically transport the pro-drug capecitabine to HER2+ breast cancer cells [96]. Otherwise, MM-302 is an HER2-targeted liposome encapsulating doxorubicin in its core, with single chain anti-HER2 antibodies (scFv) conjugated to its surface. It is already being evaluated in phase II clinical trials to treat HER2+ MBC, and is under consideration for additional oncology indications [97,98]. Finally, as regards ADCs, several examples of conjugates integrated by an anti-HER2 antibody other than trastuzumab can be encountered in the literature [89,99]. One of them is RC48-ADC, an ADC integrated by the antibody hertuzumab covalently conjugated to MMAE molecules through a cleavable dipeptide linker (hertuzumab-Val-Cit-MMAE) via cysteine residue release [100]. Its therapeutic activity against HER2+ breast carcinoma is being evaluated in phase II trials, but also some authors such as Li et al. [101] and Jiang et al. [5] have showed its efficacy as a targeted therapy for HER2+ gastric and ovarian cancers, respectively. Other examples of anti-HER2 ADCs whose antiproliferative activity is being evaluated in clinical trials for the treatment of HER2+ cancers are ARX788, TAK-522, A116, Tmab Duocarmizine, ALT-P7, DHE50815A, MEDI4276, and Tmab Deruxtecan [89,99,102,103,104,105]. More information about them, as well as about SYD98 and Tmab deruxtecan [83], can be found in Table 2.

Thus, the development of NPs, liposomes, and ADCs targeting HER2 represents a strategy of increasing interest to improve Tmab efficacy and to avoid the apparition of resistances and undesirable adverse effects. Surely, as new anti-HER2 antibodies and peptides are synthesized and their clinical administration is approved, new HER2-targeting nanosystems will emerge with enhanced therapeutic activity and reduced toxicity.

## 6. Conclusions and Future Directions

Today, it is well known that cancer is the leading cause of premature death worldwide. The conventional treatment of this complex disease involves chemotherapy, radiotherapy, and surgery, but more efficient and tolerable treatments are strongly needed to improve patient outcomes and quality of life [106].

These novel treatment strategies should be focused on the hallmarks of cancer that differentiate tumor cells from normal ones in order to reduce the apparition of side toxicology. One of such hallmarks is the overexpression of HER2 that occurs in 15–20% of breast cancers that are diagnosed, and also in other types of solid tumors, such as gastric, ovarian, or lung carcinomas [3,4,5,91]. HER2 overexpression has been associated with more aggressive tumors and a worse prognosis for patients for a long time but, since Tmab development, it has also offered a way to upgrade treatment specificity [4].

Tmab is a humanized monoclonal antibody that specifically binds one of the extracellular domains of HER2 [6]. Its clinical adjuvant administration, which is usually performed with traditional chemotherapy drugs, has positively revolutionized HER2+ breast cancer treatment since its use was approved [89,90]. Thus, nanotechnology, taking advantage of the antiproliferative activity of this antibody, has played an essential role in the production of novel HER2-guided cancer nanomedicines, and this review has focused on targeted NPs, liposomes, and ADCs. In these nanosystems, the simultaneous presence of Tmab together with a potent cytotoxic agent allows the achievement of a synergist effect that helps to reduce the needed drug dose and its secondary effects. Furthermore, acting as guided nanovehicles for chemotherapy agents, targeting NPs, liposomes, and ADCs also enhances their bioavailability, which is quite limited as a rule [10].

In order to get all these beneficial effects, the choice of a proper Tmab-anchoring strategy is crucial, especially to avoid the release of cytotoxic molecules into the circulation [17,76]. For this reason, part of the scientific community is addressing important efforts towards the development of novel covalent conjugation chemistries. In addition, with the aim of increasing Tmab efficacy and overcoming the apparition of resistances, numerous efforts are also dedicated to the synthesis of novel anti-HER2 antibodies that can be later conjugated for creating guided therapeutic nanovehicles [31,90]. As a consequence, tremendous investment is being made in this field, and increasing numbers of such nanotherapeutics are reaching clinical stages or even being commercialized in recent years [23,89,107,108].

In conclusion, it can be stated that nanotechnology holds a great promise for the apparition of combinatorial-based (drug plus antibody) cancer therapies that help to improve conventional ones, and that better manufacturing technologies enabling the synthesis of reproducible and safe systems will be fundamental in the near future [109].

## Figures and Tables

**Figure 1 nanomaterials-10-01674-f001:**
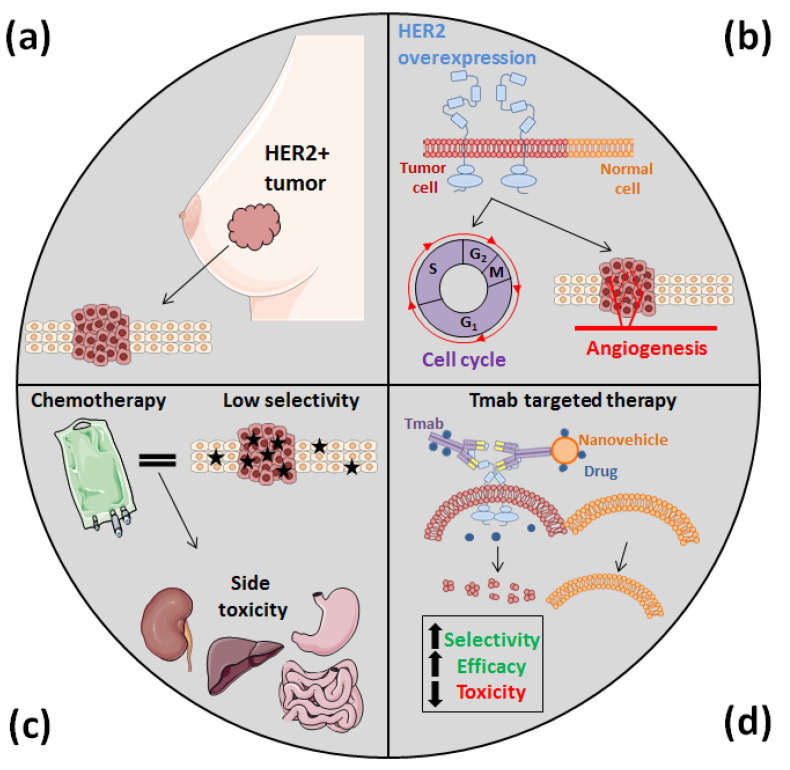
HER2 overexpression, which occurs in almost a fifth of breast cancer cases (**a**) as well as in other types of solid tumors, is related to cell proliferation and invasion and makes cancer cells more aggressive (**b**). However, this overexpression has also allowed us to develop novel nanomedicines that are more specific than conventional cytotoxic agents, which often cause acute toxicities (**c**). In the development of these new nanomedicines, since Tmab specifically recognizes HER2, it has been attached to different types of DDS, improving their efficacy and selectivity and, thus, reducing their side effects (**d**).

**Figure 2 nanomaterials-10-01674-f002:**
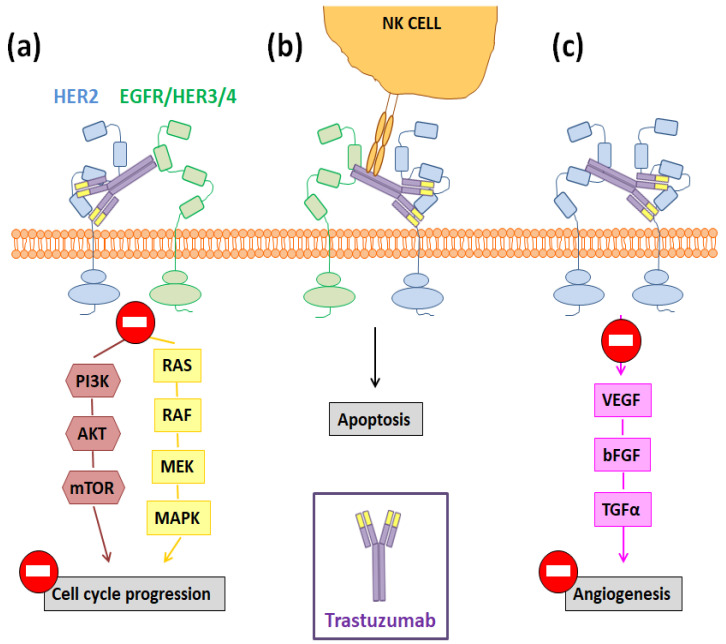
Potential molecular pathways involved in the Tmab-mediated inhibition of tumor progression once this antibody binds to the domain IV of HER2, blocking its homo- and hetero-dimerization. Such pathways are: (**a**) the inhibition of the MAPK and PI3K signaling cascades; (**b**) ADCC; (**c**) the blockage of the angiogenesis process.

**Figure 3 nanomaterials-10-01674-f003:**
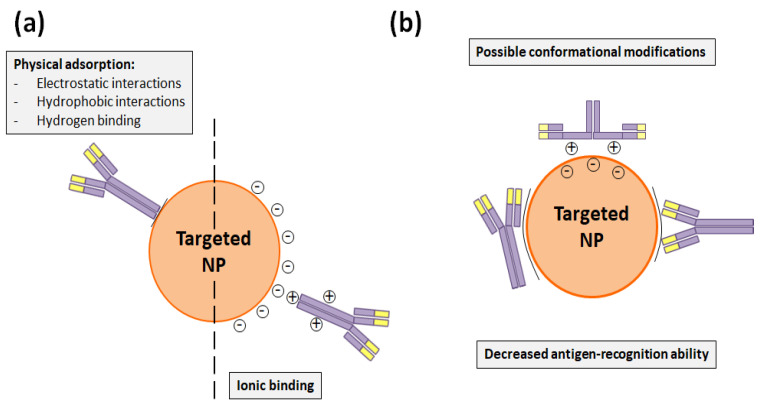
(**a**) Non-covalent binding alternatives to attach Tmab to the NP surface. (**b**) Potential conformational changes that Tmab can suffer in physical and ionic bindings and which can hinder its antigen-recognition capacity.

**Figure 4 nanomaterials-10-01674-f004:**
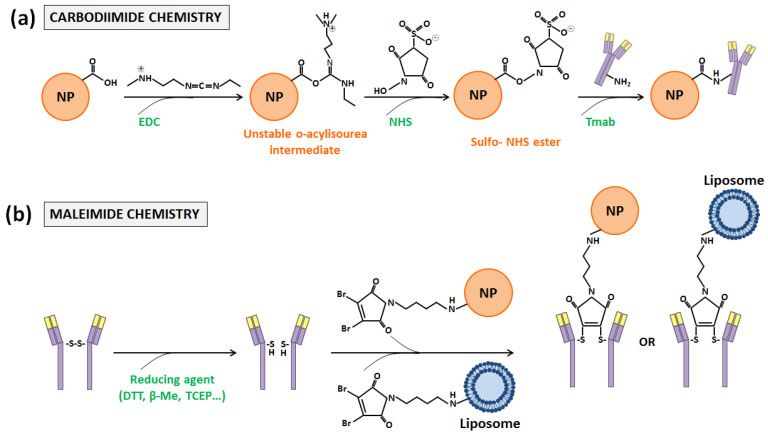
Schematic representation of the (**a**) carbodiimide and (**b**) maleimide coupling reactions between the Tmab, NP, and liposome systems.

**Figure 5 nanomaterials-10-01674-f005:**
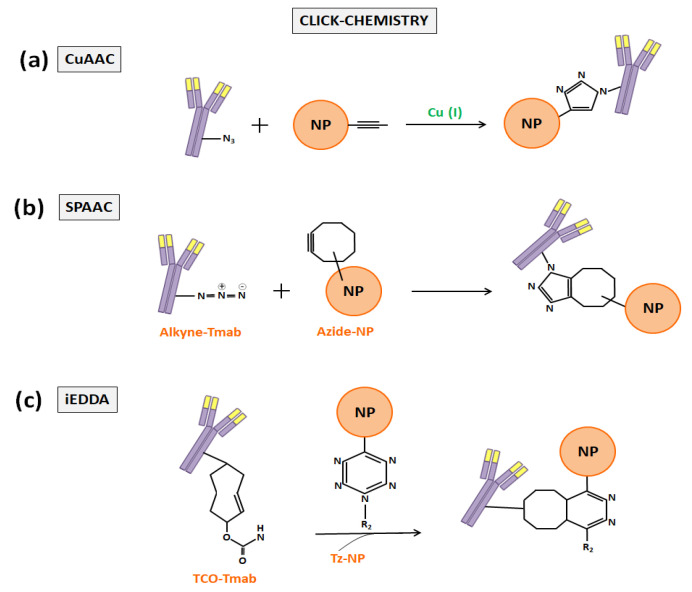
Schematic representation of the three most widely used click-chemistry reactions to anchor Tmab to NP or liposome surfaces: (**a**) CuAAC, (**b**) SPAAC, and (**c**) iEDDA.

**Figure 6 nanomaterials-10-01674-f006:**
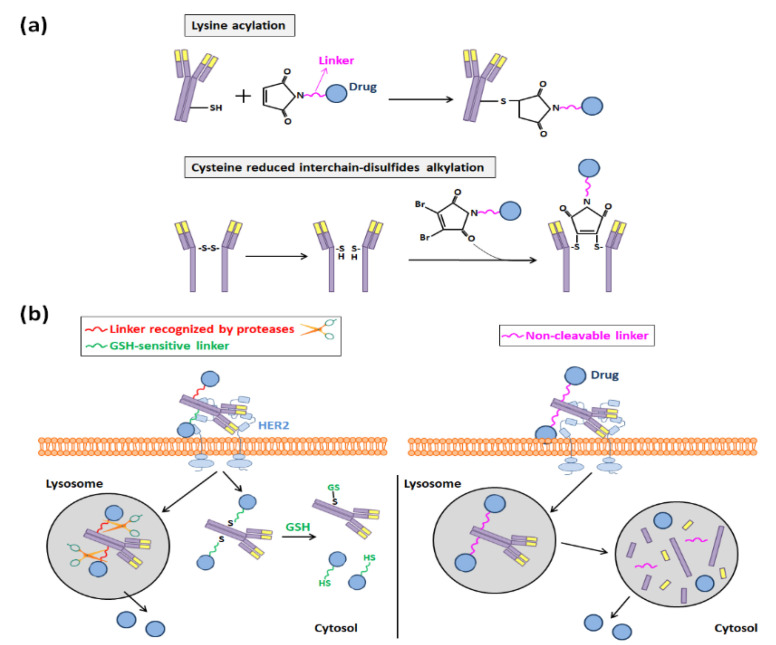
(**a**) Acylation and alkylation of lysine and cysteine residues, respectively, performed to develop Tmab-based ADCs. (**b**) Representation of the drug release that occurs in disulfide-based Tmab-based ADCs and depends on the type of linker used (cleavable vs. non-cleavable).

**Table 1 nanomaterials-10-01674-t001:** Tmab-functionalized DDS developed following different non-covalent and covalent strategies to target several sorts of HER2+ cancers.

Strategy	Type of DDS	Payload	Targeted Type of HER2+ Cancer	Reference
Physical adsorption	NPs (PLGA-TPGS)	Docetaxel	Breast	Liu et al. [39]
	NPs (PEI/PLGA)	Paclitaxel	Breast	Yu et al. [40]
	NPs (PEI/PLGA)	Docetaxel	Breast	Zhang et al. [41]
	NPs (PLGA/MTT)	Paclitaxel	Breast	Sun et al. [42]
Physical and ionic adsorption and carbodiimide chem.	NPs (PLGA)	Docetaxel	Breast	Choi et al. [43]
Carbodiimide chemistry	NPs (PLGA-Phis-PEG)	Doxorubicin	Breast	Zhou et al. [47]
	NPs (TPGS-g-chitosan)	Docetaxel	Breast	Mehata et al. [6]
	NPs (Alginate-piperazine)	Paclitaxel	Breast, ovarian	Nieto et al. [31]
	NPs (PLGA)	Cisplatin	Ovarian	Domínguez-Ríos et al. [49]
	NPs (Chitosan)	Gemcitabine	Pancreatic	Arya et al. [50]
	NPs (Magnetic)	-	Breast	Almaki et al. [53]
Maleimide chemistry	NPs (HSA)	Methotrexate	Breast	Taheri et al. [57]
	Liposomes	Rapamycin, PPɣ NPs	Breast	Nguyen et al. [58]
	Liposomes	Idarubicin	Breast	Amin et al. [59]
	DENCs	Paclitaxel, doxorubicin	Breast	Chiang et al. [60]
	Liposomes	Magnetic NPs	Breast	Jang et al. [61]
	NPs (Polyamidoamine dendrimers)	Cisplatin	Ovarian	Kesavan et al. [62]
	NPs (HSA)	-	Anyone	Steinhauser et al. [63]
	NPs (Gold)	-	Gastric	Kubota et al. [64]
Click-chemistry (SPAAC)	NPs (PLGA)	-	Breast	Greene et al. [66]
Click-chemistry (iEDDA)	Liposomes	SN38	Anyone	Yoo et al. [72]
	-	Fluorine-18	Breast	Keinänen et al. [73]

**Table 2 nanomaterials-10-01674-t002:** Examples of novel anti-HER2 ADCs, different from the well-known T-DM1, whose efficacy and safety are already being evaluated in clinical trials.

ADC	IgG	Payload	Clinical Trial Phase ^5^	Indication	Developer
RC48-ADC ^2–4^	IgG1 (Hertuzumab)	MMAE	Phase I	Solid tumors	Regemen
ARX788 ^2,3,6^	Engineered IgG1	MMAF	Phase I	Breast, stomach cancers	Ambrx
TAK-522 (XMT-1522) ^2,3,6^	IgG1 (HT19)	AF-HPA	Phase I	NSCLC, breast, gastric cancers	Mersana
A116 ^2^	Not disclosed	Not disclosed	Phase I	Breast cancer	Klus Pharma
Tmab Duocarmazine (SYD985) ^1–3,6^	IgG1	Seco-DUBA	Phase II	Endometrial cancer	Synthon
ALT-P7 ^2^	IgG1 (HM2, Tmab biobetter)	MMAE	Phase I	Breast cancer	Altrogen
DHES0815A ^2^	IgG1	PBD-MA	Phase I	Breast cancer	Genentech
MEDI4276 ^2,3^	Bi-specific IgG1 (Tmab ScFv)	AZ13599185	Phase I/II	Breast, gastric cancers	MedImmune
Tmab Deruxtecan (DS-8201a) ^1–3,6^	IgG1	DXd	Phase II	Breast cancer	Daiichi Sankyo

^1^ [83], ^2^ [89], ^3^ [99], ^4^ [100], ^5^ [104], ^6^ [105]. AF-HPA: auristatin F-hydroxypropylamide; NSCLC: non-small cell lung carcinoma; PBD-MA: pyrrolo[2,1-*c*][1,4]benzodiazepine monoamide.

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
