# Peer review of "Trastuzumab: More than a Guide in HER2-Positive Cancer Nanomedicine"

_nanomaterials, 2020, doi:10.3390/nano10091674_

Round 1

Reviewer 1 Report

The author here presents a nice summary on Trastuzumab. The work may be accepted after minor spell checks.

Author Response

The author here presents a nice summary on Trastuzumab. The work may be accepted after minor spell checks.

Thank you so much. The manuscript has been checked by a native English-speaking person and the comments of the other reviewers have also been taken into account to correct spell errors. Manuscript modifications have been highlighted in red.

Reviewer 2 Report

The paper from Dr. Nieto et al presents a comprehensive review of the mechanisms and beneficial aspects of trastuzumab incorporation in nanoparticles for the treatment of HER+ breast cancer.

The paper is very well written and includes elucidative pictures that help the reader summarizing the information. Minor points would improve the manuscript:

1 - Figure 1 by itself is not self-explanatory. The legend should be extended to allow interpretation.

2 - Line 299: remove “had been”

3 - Line 347: replace “cleavage” by “cleaved”

Author Response

The paper from Nieto et al. presents a comprehensive review of the mechanisms and beneficial aspects of trastuzumab incorporation in nanoparticles for the treatment of HER2+ breast cancer.

The paper is very well written and includes elucidative pictures that help the reader summarizing the information. Minor points would improve the manuscript:

Thank you so much for all your comments.

  1. Figure 1 by itself is not self-explanatory. The legend should be extended to allow interpretation.

“Bullets” in Figure 1 have been numbered and the legend has been modified and extended.

  1. Line 299: remove “had been”.

Thank you for the spell correction. It has been taken into account and correct in line 310 (previously 299).

  1. Line 347: replace “cleavage” by “cleaved”.

Thank you again. The replacement has been done in line 358 (previously 347).

Reviewer 3 Report

The review is very interesting.

  • However, the English in whole document should be extensively improved.
  • The sentence in lines – 382-383 should be corrected: “As a result, 3.5 molecules of the drug were incorporated per antibody, ensuring a good aqueous solubility of T-DM1” according to reference 82 . In ref. 82 the surname of the 2nd author is Chary and not Char.
  • The word “incorporated” along the manuscript is sometimes used to refer that Tmab or other antibodies are decorated at nanoparticles surface. Do you think the word “incorporated” is adequate ?
  • The legend of Figure 1 should be reformulated
  • In Figure 4 the word “lysosome” should be changed to “liposome”
  • In lines 294-295 - “After that, preferred copper-free click chemistry was developed, based on cycloadditions between strain-promoted alkynes and  azides (SPAAC) [68]” Please clarify this sentence.
  • In Table 1. What does it mean “HAS”? Please correct. An uniformity in the classification of the types of DDS should be carried out. For instance, it appears pegylated liposomes, liposomes, or DPPC-DSPE-PEG; PLGA NPs or PLGA; Kubota et al, is reference number 64 not 65.
  • Table 2 should be reviewed and improved .
  • In section 6 – lines 511 - 515 – the word “lysosomes” should be replaced by liposomes
  • the meaning of all abbreviations should be included in the manuscript

Author Response

The review is very interesting. However, the English in whole document should be extensively improved.

Thank you so much. The manuscript has been now checked by a native English-speaking person. All modifications have been highlighted in red.

  • The sentence in lines – 382-383 should be corrected: “As a result, 3.5 molecules of the drug were incorporated per antibody, ensuring a good aqueous solubility of T-DM1” according to reference 82. In ref. 82 the surname of the 2ndauthor is Chary and not Char.

The mentioned sentence (now in lines 393-395) has been modified. In addition, the surname of the second author in reference 82 has been corrected.

  • The word “incorporated” along the manuscript is sometimes used to refer that Tmab or other antibodies are decorated at nanoparticles surface. Do you think the word “incorporated” is adequate?

The words “incorporate/incorporated” have been replaced by “load/loaded” and “functionalize/functionalized with” along the manuscript (lines 155, 171, 257, 269, 375 and 478).

  • The legend of Figure 1 should be reformulated.

“Bullets” in Figure 1 have been numbered and the legend has been modified.

  • In Figure 4 the word “lysosome” should be changed to “liposome”.

Thank you for the correction, it has been taken into account.

  • In lines 294-295 - “After that, preferred copper-free click chemistry was developed, based on cycloadditions between strain-promoted alkynes and azides (SPAAC) [68]” Please clarify this sentence.

Changes have been made in lines 302-306 in order to improve the comprehension of this paragraph.

  • In Table 1, what does it mean “HAS”? Please correct. Uniformity in the classification of the types of DDS should be carried out. For instance, it appears pegylated liposomes, liposomes, or DPPC-DSPE-PEG; PLGA NPs or PLGA; Kubota et al, is reference number 64 not 65.

“HAS” should be “HSA”, which is the abbreviation corresponding to Human Serum Albumin (now included in the abbreviation section at the final of the manuscript). This typing error has been corrected and the referencing number of Kubota et al. has been changed, too. Moreover, DDS classification in Table 1 has been standardized, firstly mentioning the type of DDS (NP or liposome) and, between brackets, the main composition of the nanoparticles (NPs) (PLGA, PEI/PLGA, chitosan…).

  • Table 2 should be reviewed and improved.

Table 2 has been extended and all the anti-HER2 ADCs which are now in clinical trials have been included. Data have been updated and two more references (105 and 106) have been added.

  • In section 6 – lines 511 - 515 – the word “lysosomes” should be replaced by “liposomes”.

Thank you again for the spell corrections. “Lysosomes” has been replaced by “liposomes” in lines 525 and 528 (previously 511 and 515 lines).

  • The meaning of all abbreviations should be included in the manuscript.

The meaning of all abbreviations, alphabetically sorted, has been included at the final of the manuscript, before the statements of author contributions, funding and conflicts of interest.

Round 2

Reviewer 3 Report

Sorry but I disagree with the words load/loaded to refer that Tamb is attached/anchored/ binded at NPs or liposomes surface. In my opinion loaded or incorporated means that a compound or molecule is in the inner of the delivery system and not at their surface.

In table 2. each example included, must be associated
to a reference.

Author Response

Sorry but I disagree with the words load/loaded to refer that Tmab is attached/anchored/ bound at NP or liposome surface. In my opinion loaded or incorporated means that a compound or molecule is in the inner of the delivery system and not at their surface.

Thank you for your correction. The words “load/loaded” that referred to Tmab attachment to NP and liposome surface have been replaced by “to bind/bound”, “attached” or “anchored”. Modifications can be found in lines 155, 269, 367, 383, 439 and 530 of the manuscript, as well as in the legend of the Table 1.

In Table 2 each example included must be associated to a reference.

A couple of additional references have been included in Table 2 and, now, each anti-HER2 ADC is associated with the corresponding references through cardinal numbers.
